# Balancing the uncertain and unpredictable nature of possible zoonotic disease transmission with the value placed on animals: Findings from a qualitative study in Guinea

**Tilly A. Gurman**[1]*, **Kendela Diallo**[2], **Elizabeth Larson**[1], **Kathryn Sugg**[1], **Natalie Tibbels**[1]

**1** Johns Hopkins Center for Communication Programs, Baltimore, Maryland, United States of America,
**2** Johns Hopkins Center for Communication Programs, Conakry, Guinea

* tgurman@jhu.edu

**Data Availability Statement:** All data are publicly available within the Qualitative Data Repository (https://doi.org/10.5064/F6HURDNS.V1). Gurman,

## Abstract

Zoonoses, or diseases that pass between animals and humans, represent a major threat to global health and global economies. In Guinea, zoonotic diseases (e.g. rabies, Lassa fever) have been at the forefront due to recent outbreaks and government priorities. Much like many other diseases, zoonotic disease prevention demands a thorough and culturally nuanced understanding of the factors that influence preventive behaviors. To gain this knowledge and enhance risk communication for priority zoonotic diseases, this qualitative study conducted focus group discussions, in-depth interviews, and observations in three Guinean prefectures. Study participants included individuals who interact with animals or influence human-animal interactions, (e.g., veterinarians, local leaders, human health providers, butchers, hunters, general population). A total of 229 individuals participated in the study. Data analysis, which combined deductive and inductive coding, found that although individuals generally had basic knowledge about zoonotic diseases, a gap existed between knowledge and practice. In exploring possible reasons behind this gap, several key themes arose, the two most novel being the focus of this paper. First, participants described living in an uncertain world where they lack control over the behaviors of others. Many participants described uncertainty over the vaccine status of stray dogs or even those of their neighbors, making them feel powerless over rabies. Second, animals serve as a main source of livelihood (income, investment, or savings) for individuals. The value placed on livestock may, in turn, drive and impede prevention behaviors such as vaccinating animals or avoiding the sale of unsafe meat. Given that the Guinean government's list of priority zoonotic diseases continues to evolve, the need to discover ways to effectively promote multiple related prevention behaviors remains pertinent. The insights from this study can inform existing and future programs for the prevention, control, and surveillance of zoonotic disease in Guinea and other similar countries.

Tilly, A.; Diallo, Kendela; Tibbels, Natalie. 2024. "Data for: 'Balancing the uncertain and unpredictable nature of possible zoonotic disease transmission with the value placed on animals: Findings from a qualitative study in Guinea'". Qualitative Data Repository. https://doi.org/10.5064/F6HURDNS.V1.

**Funding:** Funding for this study came from the Global Health Security Administration at the United States Agency for International Development (USAID #126722). TG, KD, EL, KS, and NT all received some salary support from the funder. The specific roles of these authors are articulated in the 'author contributions' section. The funder informed the study design by providing suggestions about which zoonotic diseases were of interest in Guinea. No additional external funding was received for this study. The funder had no additional role in study design, data collection and analysis, decision to publish, or preparation of the manuscript.

**Competing interests:** The authors have declared that no competing interests exist.

## Introduction

Zoonoses, or diseases that pass between animals and humans, represent a major threat to global health. Current estimates indicate that around 75% of all emerging infectious diseases are zoonotic [1]. The impact of zoonotic disease on the human population is substantial, with a 2012 report from the International Livestock Research Institute determining that the 56 most common zoonotic diseases were responsible for 2.7 million deaths and almost 2.5 billion illnesses globally [2]. Rabies alone, for example, is responsible for approximately 59,000 deaths annually and 3.7 million disability adjusted life years lost [3]. Zoonoses also cause severe economic consequences for the global community. The World Bank estimated that between 1997 and 2009, six pandemics–Nipah virus in Malaysia, West Nile fever in the United States of America (US), severe acute respiratory syndrome (SARS) in Asia and Canada, highly pathogenic avian influenza (HPAI) in Asia and Europe, bovine spongiform encephalopathy (BSE) in the United States and the United Kingdom, and the Rift Valley fever (RVF) in East Africa–resulted in $80 billion in losses [4]. Most recently, the severe acute respiratory syndrome coronavirus 2 (COVID-19) pandemic, caused by the SARS CoV-2 virus, has best demonstrated the threat of zoonoses. As of March 2023, COVID-19 has killed over 6.8 million people globally and infected well over 676 million people [5]. In addition to COVID-19, many of the most well-known, and consequential, infectious diseases are zoonotic, including the human immunodeficiency virus (HIV), tuberculosis, rabies, and the Ebola virus disease (EVD).

Recently, diseases such as rabies, Lassa fever, and HPAI have also caused concern in the West and Central African regions. To curb the impact of these diseases, international and country-based organizations have invested significant resources in preparedness and response to zoonotic disease. However, while regional, national, and international actors play important roles, those at the community level also hold a key position in prevention and response [6]. Moreover, a variety of individual behaviors can prevent zoonotic events or zoonotic disease transmission, such as vaccination, good hygiene practices during food preparation, care-seeking for zoonotic diseases, or wearing protective equipment when in contact with live animals or animal products [7, 8]. Individual and community level factors influence these types of behaviors, including knowledge or misconceptions about how the disease is transmitted or treated, beliefs, or perceived norms [9]. Individual behavior particularly drives disease risk and transmission for zoonotic diseases in resource-limited settings where human-animal interactions are pervasive and urgent economic realities override motivations to avoid more distant threats [10]. Community members are the primary caretakers of many of the animals that transmit zoonotic disease, and given the proper resources, can inhibit the pathway of transmission. As a result, the successful prevention and management of zoonotic diseases demands a thorough and culturally nuanced understanding of the various factors that influence human behavior at multiple levels.

The West African nation of Guinea has been hard-hit by zoonotic disease outbreaks in recent years. Guinea is one of the poorest countries in the world, with a per capita GDP just over 1,000 USD in 2019 [11]. According to the World Bank, 43.7% of the Guinean population lives below the national poverty line during 2018–2019 [12]. Raising livestock is the second most important activity in the rural sector. The Strategy and Development Office of the Ministry of Livestock estimated that the livestock population in 2016 was 6,759,000 cattle, 2,380,000 sheep, 2,851,000 goats, 130,000 pigs, and 30,000,000 poultry and other captive birds. Raising livestock provides the livelihood of 30% of the rural population [13].

The flow of commerce between different cities and villages is a risk factor for transmission of zoonotic diseases, as farmers transport their livestock to the markets of major cities. Conakry is a densely populated and diverse urban center, whereas Kankan and Nzérékoré have

both urban and rural areas. Conakry, as the capital and city with the largest population, is the arrival destination for the majority of animals from other parts of the country, including Kankan and N'Zérékoré. and the city with the largest number of consumers. Conakry, however, has no modern slaughterhouses [14]. The prefecture of Kankan is the capital of the upper Guinea region and is an area of high livestock production and consumption [14]. Kankan's economy is based on agriculture and livestock, although some of the meat consumed there comes from hunting. Meanwhile, N'Zérékoré is a densely populated prefecture in the forested region of Guinea, with hunting and consumption of bushmeat common throughout. The area is simultaneously home to numerous pig breeders.

Conakry, N'Zérékoré, and Kankan have been of concern to the Guinean government for zoonotic diseases. At the time of the study, Kankan had an elevated risk for yellow fever and brucellosis, while Nzérékoré and Conakry had an elevated risk for yellow fever, rabies, and anthrax. The context regarding both animal consumption and zoonotic disease in Conakry, N'Zérékoré, and Kankan highlighted above underline the importance of better understanding risk around zoonotic transmission and opportunities for intervention in these areas.

The threat of zoonotic diseases in West and Central Africa came to the forefront in recent years with several outbreaks of EVD, including the 2014–2016 epidemic that led to 3,814 cases and 2,544 deaths in Guinea [15]. Still, to date, there has been limited research focused on the factors influencing prevention behaviors related to zoonoses. Most current information comes from the EVD epidemic. For example, one recent study examined the relationship between socioeconomics, food security, and public health, and determined that people in rural settings faced numerous cultural and economic constraints that kept them from accessing other meat than bushmeat [16]. A preliminary review of the literature for other zoonotic diseases in Guinea suggests that little research exists on individual, sociocultural, and structural factors influencing human-animal interactions in the country. Likewise, few research studies have examined motivators and barriers to uptake of zoonotic disease prevention behaviors. A literature review of individual and social risk factors for zoonotic diseases across West Africa highlights the need to explore preventive behaviors such as cooking meat thoroughly, boiling milk, vaccinating pets and/or livestock, wearing protective clothing, and washing hands with soap after contact with animal carcasses, aborted fetuses, or amniotic/vaginal fluid from affected animals that may affect multiple diseases [17].

In response to the dearth of information on preventative behaviors and zoonoses in Guinea, the Global Health Security Agenda (GHSA) worked to identify zoonotic diseases that would be prioritized for investments in surveillance, preparedness, and response in the country. The GHSA represents a global cross-sectorial network aimed at improving preparedness and response to infectious diseases worldwide [18]. GHSA collaborated with government stakeholders and non-government organization (NGO) partners in Guinea to consider various diseases, the frequency of occurrence and epidemic potential, and identified anthrax, rabies, brucellosis, HPAI, Lassa fever, and other viral hemorrhagic fevers, such as EVD and RVF as priority diseases. The current study aimed to unearth factors influencing prevention behaviors related to these zoonotic diseases in order to inform current and future policy and programmatic efforts in Guinea as well as other countries. An added benefit of this exploration of factors that influence zoonotic disease prevention behaviors is to help fill the above-stated gap in the literature.

## Methods

### Study design

Researchers used three qualitative methods to understand different aspects of zoonotic disease prevention behaviors in the Guinean prefectures of Conakry, N'zérékoré, and Kankan. First,

**Table 1. Prevention behaviors explored during focus group discussions about zoonotic disease perceptions in Guinea.**

| Prevention behaviors |
| --- |
| • Keep animals separate from living areas |
| • Disinfect animal pens |
| • Avoid dog bites |
| • Seek immediate care at health center for dog bites |
| • Vaccinate animals |
| • Boil milk for 30 minutes before drinking |
| • Sterilize knives and surfaces used to cut fresh meat |
| • Cook meat well, only eat meat that is well cooked |
| • Avoid eating meat from sick animals |
| • Avoid eating bushmeat |
| • Cover cuts or wounds on the skin when handling animals |
| • Wear protective clothing while touching carcasses |
| • Bury sick animal carcasses and aborted fetuses |
| • Avoid eating fruit already partly consumed by an animal |
| • Store food in covered containers to protect it from rodents |

in-depth interviews explored individuals' interactions with animals, their awareness of zoonotic diseases, their perceived role in an epidemic response, and how health information flows within Guinea for the three respective prefectures. Second, focus group discussions explored community norms and attitudes toward animals, zoonotic diseases, and the desired prevention behaviors (see Table 1), as well as preferred information channels. During the focus group discussions participants engaged in a pile-sorting activity about various prevention behaviors. Facilitators used a pile of illustrated cards, with each card depicting a specific zoonotic disease prevention behavior. The facilitator then asked participants to assess the behavior on each card according to two dimensions, response efficacy and feasibility. First, participants assessed response efficacy in terms of how effective the behavior is at preventing disease (effective, more or less effective, or not effective at all). Then, participants assessed the feasibility of performing the behavior (easy to do, somewhat easy to do, or difficult to do). Facilitators promoted discussion and debate to encourage a variety of opinions. The activity culminated in participants voting, with majority decision, as to where to classify each behavior along both dimensions.

Finally, researchers conducted direct observations, which is a qualitative data collection technique where researchers join individuals in a specific environment doing activities of interest and observe without participating [19]. Members of the research team joined participants during one workday and documented through notetaking and photographs any factors in the physical environment that influenced human-animal interactions.

## Study sample

The data collection team conducted purposeful sampling, selecting adult participants based on their relevance to the research questions, specifically those who interact with animals or influence those interactions. In-depth interviews engaged local community leaders such as imams or neighborhood chiefs, media professionals, health providers (formal or community-based) for both humans and animals, including formally trained veterinarians. Members of the general population—stratified by gender and setting (urban/rural)—and professional animal handlers *[Manutentionnaires]* including livestock sellers, hunters, and butchers, and animal

farmers/breeder participated in focus groups. Candidates for participant observation were animal handlers or staff at veterinary offices that had already participated in either a focus group discussion or in-depth interview.

Members of local associations/ groups—identified in collaboration with executives of the Ministry of Livestock through its Directorate of Veterinary Services—recruited eligible study participants and assisted organizing study activities. Local neighborhood chiefs identified potential participants from the general population and were charged with recruiting for two urban and two rural focus group discussions with members of the general population. Leaders of the livestock associations or other relevant local officials identified potential participants in the other categories and recruited health workers, media professionals, community leaders, and people who interact with animals professionally. Participants were included if they were aged 18 or older (general population) or part of one of the sub-populations of interest described above; they were excluded if they lacked the capacity to provide consent or were unable to communicate in French or one of the four local languages spoken by the data collectors.

## Data collection

Data collectors consisted of male and female Guinean researchers with experience conducting qualitative research in at least one of the four common local languages in the study sites (Malinke, Soussou, Poular, or Kpelle). Data collectors participated in a multi-day training during which they reviewed and practiced with study instruments, including the interview, focus group, and participant observation guides as well as the informed consent scripts (for interview and focus group guides, see S1 and S2 Appendices). Focus groups with the general population were separated by gender and matched with same-gender facilitators. Men and women participated together in focus groups with people who worked professionally with animals, given that the guides focused on occupational (not household-level) perspectives and that we anticipated difficulty recruiting enough women in the occupational categories.

All participants provided written consent before participating in the study. Focus groups and interviews took place at a location convenient for the participants (e.g. community center, church, school) while ensuring privacy. Observations occurred at work sites such as farms, veterinary offices, livestock markets, or slaughterhouses. Data collectors observed the participants interacting with animals and took notes using a template, as well as photographs. Interviews and focus groups were conducted in French or one of the local languages using a semi-structured interview or focus group guide. Interviews included questions about the participant's daily interactions with animals and perceived risk of zoonotic disease, as well as inquiring about their role in an epidemic response and questions about the flow of health information in Guinea. Focus groups explored normative practices around interactions with animals and animal hygiene and sought to better understand which sources of health information were considered trustworthy. Focus group discussion guides framed questions around what participants perceived that other members of their community would do or believe, rather than what the participants themselves would do or believe. (See S1 and S2 Appendices for guides.) Facilitators were trained on managing power dynamics in group discussions to encourage all participants to speak freely. All activities were audio-recorded. Each participant only participated in one interview or focus group. On average, the focus groups lasted around two hours and interviews lasted 45 minutes, with participant observations lasting between 4–8 hours.

Data collection lasted approximately two weeks during September 2019. Data collectors then simultaneously transcribed and translated the interviews and focus groups from the local

language into French. All transcripts were validated by spot checking for quality and accuracy (listening to two minutes of audio for every 20 minutes of recording). Discrepancies led to a review where the transcriber listened to the full audio and revised the transcript. Quotations included in this manuscript were translated into English by two bilingual authors (TG and NT).

The research protocol, guides, and consent forms were approved by the [institution anonymized] Institutional Review Board [IRB#9754] and the Guinean national research ethics committee (Comité National d'Ethique pour la Recherche en Santé).

## Data analysis

TG and NT developed a coding framework based on a literature review of zoonotic diseases and prior qualitative research conducted in West Africa. Four data collectors and their supervisor along with the study manager (KD) imported and conducted deductive coding with all transcripts in Atlas.ti (ATLAS.ti GmbH, Berlin, Germany), double coding 14% of transcripts and discussing any discrepancies to achieve consistency.

The study's principal investigator (TG) facilitated five-days of collaborative data analysis. This participatory data analysis workshop gathered 12 participants including several data collectors, the study manager (KD), staff from the project that funded this research, as well as representatives from various government ministries. The participatory approach to analysis corresponds to a framework analysis methodology, with the data familiarization and framework identification partially done in advance [20]. During the workshop, participants reviewed selections of the transcripts that corresponded to the deductive coding framework and worked in small groups to identify themes via inductive coding. Each small group worked with pre-assigned data from either Conakry, Kankan, or N''Zerékoré. Once the small groups finalized the themes that emerged from their prefecture's data, they presented them to the larger group. The entire team then collaboratively generated themes that cut across multiple preventive behaviors and charted them with illustrative quotations. During the data analysis process, saturation was reached given that the themes were recurring across the three prefectures.

## Results

Across the three prefectures, the team conducted 24 focus groups (with 205 participants) and 24 interviews, as well as 15 participant observations (see Table 2). The participants were, for the most part, equally divided across the three prefectures. There were eight in-depth interviews per prefecture and five participant observations conducted per prefecture. Of the eight focus groups conducted per prefecture, the total number of participants ranged from seven to ten per group. The total number of focus group participants was 61 in Conakry, 75 in Kankan, and 69 in N'Zérékoré. A total of 229 individuals participated in the study, of which only 58 were women.

Study data indicated that individuals generally had basic knowledge about zoonotic diseases. Participants discussed the link between certain behaviors, disease exposure, and risk of infection. At the same time, a gap existed between the knowledge of prevention behaviors expressed by participants and them practicing those behaviors. In trying to understand some of the factors behind this gap, seven themes emerged during the data analysis workshop that cut across the various preventive behaviors of interest (see Table 3). The authors then prioritized two novel themes which they deemed as key to understanding some of the nuanced influences on zoonotic disease prevention behaviors.

Table 2. Participants of qualitative study regarding perceptions of zoonotic disease in Guinea, by type of data collection method.

| TYPE OF PARTICIPANT | | DATA COLLECTION METHOD | | |
|---|---|---|---|---|
| | | IN-DEPTH INTERVIEW | FOCUS GROUP | OBSERVATION |
| Health Professional | Human health providers | 6 | | |
| | Veterinarians | 3 | | 3 |
| Community Stakeholder | Community leaders | 12 | | |
| | Media professionals | 3 | | |
| Animal Handlers | Butchers | | 30 | 4 |
| | Animal farmers/ breeders | | 19 | 5 |
| | Vendors | | 32 | 3 |
| | Hunters/ transporters | | 14 | |
| General Population | Women (urban) | | 34 | |
| | Women (rural) | | 19 | |
| | Men (urban) | | 37 | |
| | Men (rural) | | 20 | |
| | Total individuals | 24 | 205 | 15 |

## Recurring uncertainty and lack of control

Participants expressed lack of control over the behaviors of others such as whether people vaccinate their dogs against rabies, or whether they could trust the meat they purchased from butchers. This reduced autonomy in relation to perceived ability to control risk of zoonotic disease led to uncertainty around whether individual actions would be sufficient in preventing infection.

First, participants across all three prefectures described a reality in which their daily lives are full of uncertainty and ambiguity regarding the health status of animals. The uncertainty and lack of control participants described may manifest in multiple ways, including not knowing whether people vaccinate their pets for rabies, neighbors letting their pets run loose, and the routine presence of stray animals. One woman in urban N'zérékoré commented about how people in her community raise their dogs, *"They don't take care of them. They don't vaccinate them, and these dogs are abandoned and left to fend for themselves. They are not tied up and if they manage to bite you, it becomes complicated."*

Table 3. Themes from qualitative study on perceptions of zoonotic disease in Guinea.

| | |
|---|---|
| 1 | Although people may have general basic knowledge and awareness of zoonotic diseases, they often fail to put knowledge into practice. |
| 2 | People's traditional and religious beliefs, existing habits, and norms in their families and community may influence their ability to engage in prevention behaviors. |
| 3 | Veterinarians play a vital role in the control and prevention of zoonotic diseases, including in the diffusion of information. |
| 4 | Access constraints and structural barriers prevent people from engaging in healthy behaviors. |
| 5 | People rely on multiple sources and channels of information, which can be an asset for the dissemination of information during emergencies. |
| 6* | People live in a world with a lot of uncertainty, which can make it difficult to adopt effective prevention behaviors. |
| 7* | People place great value on their animals for their livelihood, which influences their behaviors. |

* = Novel themes highlighted in this article

A woman in urban Conakry similarly remarked, *"you don't know what time the owner of the dog releases his dog, and you can't know when you can meet. So you can't take precautions because you don't know when he can come, you don't know when the owner will release him."*
A male focus group participant from urban Conakry concurred:

*You can prevent disease, but it is difficult to avoid a dog left to its own devices. There is no way to avoid them. It is also not easy for individuals to take initiatives to vaccinate stray dogs.*

A veterinarian worker from Kankan expressed that his concern with rabies is that:

*"as the system of breeding makes that people do not manage to keep the animals well. And when an animal gets sick, instead of staying at home, the animal keeps biting everything in its path. Cattle, rocks, whatever it meets on the way. So it's very dangerous. And it's an irreversible disease. When it breaks out, it's over, it's irreversible."*

Participants characterized vaccination as a way to limit the consequences of a dog bite but expressed frustration that there was often uncertainty about a dog's vaccination status. Participants described that people are sometimes dishonest about dog ownership or vaccination status to avoid being held responsible. These beliefs were augmented by the notion that avoiding dog bites was a difficult behavior given the pervasive presence of stray dogs. A man in urban Kankan described what he saw as the only way to resolve this presence, stating. *". . . a girl went to get some water at the riverbank, a dog came from behind and bit her. Could she have avoided this accident*? *No, because the dog bit her by surprise. If you want to prevent that, you have to kill all the dogs in the area."*
In addition to protecting human health, vaccination was accepted as an important part of animal health. Participants felt that trusted influencers (e.g., local leaders, veterinarians) advocate for the utility of animal vaccination. Despite certain rumors circulating about the negative effect of vaccines on animal strength and health, participants disapproved of community members who refused to vaccinate; some participants even recommended euthanizing unvaccinated dogs to increase motivation. Only half of the focus groups deemed vaccination as feasible, largely due to cost and access.
Much uncertainty also lay around the safety of the meat individuals purchase and consume, leading to a lack of trust in meat which ultimately might lead people to avoid eating certain meat altogether. For example, an animal handler from Kankan stated, *"When I doubt an animal, I don't eat the meat even if I don't know the nature of the disease"*. Participants described not knowing the health status of animal before slaughter, the type of meat, or how well it was cooked when eating outside the home, such as while visiting a friend's house for dinner. For example, a man from urban Conakry commented about avoiding eating meat from a sick animal:

*For me, it's not easy, because someone can kill a sick animal, and you who come to buy, you don't know. You buy. You send it home. You prepare it. And if it is badly cooked, if you eat it, you contract the disease. Or even if you prepare it well, but if the animal has been sick, you contract the disease too. So really, it's not easy at all, because we don't know where the animal was killed or if it was vaccinated or not with the veterinarians.*

An animal handler from N'zérékoré similarly described,

*There are some butchers who slaughter certain animals to sell them, even though they know that the animal cannot live for long, but that does not prevent you from slaughtering it. You, the consumer, who doesn't know anything about it, comes and pays for this kind of meat and consumes it.*

To counteract these various challenges around the uncertainty of the quality of meat, an animal seller from Conakry expressed his desire for improved oversight and assistance from veterinarians, stating,

*When you send the animals to be slaughtered, you're not there. You don't know the state in which these animals arrived. Were they sick or not before they came to the slaughterhouse? You don't know anything. You only see the meat. What would help the population in this situation is the assistance of veterinarians. They do their checks before and after they slaughter the cow, before the meat is delivered to the market. Otherwise, we'll eat it and. . . right now all diseases come from food.*

Participants also highlighted the challenges to determining what type of meat is being sold —whether bushmeat or livestock. A man from urban Conakry assessed the complexity of avoiding eating bushmeat, saying, *"It is more or less difficult because there is no bush here. But if I buy meat from a travelling salesman, I can't find out the origin of the meat."* Another man added, *"If I visit someone and he hands me good meat I cannot verify its origin."*

Regarding the consumption of bushmeat, a man from an urban area of N'zérékoré similarly highlighted, *"It's not easy, it's not easy because just now we go to the market, there are cured meats. The women sell any kind of meat. We don't know what kind of meat it is"*. At the same time, some beliefs encourage the consumption of bushmeat, as discussed by a rural man from Kankan, *"The meat of some bush animals enhances our health. When you get ''sé', which is a disease that turns the urine yellow, it is cured by the meat of partridges. Just like red monkey meat."*

Uncertainty around the hygienic practices of people slaughtering animals for meat consumption also surfaced. For example, when talking about the behavior of sterilizing cutting surfaces and utensils, a hunter from Kankan stated, *"People cut the animals' throats without cleaning the knife before putting it in the sheath just as many women can also take up to 3 days without washing their knives. These are the consequences of negligence and lack of control."*

In some situations, people may have to use or borrow materials from others who may or may not practice the same level of hygiene. One urban woman in Conakry lamented when assessing the feasibility of sterilizing knives and cooking surfaces, *"But typically it's not one person who actually uses it [a knife] . . . you're getting ready, you're in a hurry, but you don't see your knife. So, you have to go to your neighbor's house and borrow her knife real quick. See? So it's not easy."*

In an attempt to better navigate uncertainty, participants in Conakry, in particular, described the need to be able to evaluate in real time health-related information shared by local authorities. For example, a human health provider from Conakry described, *"When I first hear a piece of information, I try to analyze it, to see to what extent the information is reliable, and then I try to pass it on. . ."*.

A local leader in Conakry also highlighted the importance of the dissemination of quality information by authorities during moments of crises. When asked to describe a recent zoonotic-related event where they had a role communicating information sent by authorities to their community, the leader explained,

*People reacted positively because information is the source of nourishment when you don't have good information you fall into a hole, but when you have good information you can feel happy, so the information came from the central office and we took this information to the community level and each member of the community took care to listen to us and to follow the recommendations given to this effect.*

It should be noted, however, that the navigation of and access to improved information was less often discussed in N'zérékoré and Kankan.

## Importance of animals for one's livelihood

Across the three prefectures, participants shared that for many people in Guinea, their animals are their main source of income, investment, or savings. A man from urban Conakry offered examples of why people might not call the veterinarian when their animal is sick, stating:

*The reasons for raising animals are not the same. Some raise a sheep for example to make a sacrifice. When this animal falls ill, this farmer will try to treat it. Others raise animals for their own consumption, so when these animals get sick, they kill them and consume them immediately. Others raise animals for commercialization. So when they get sick, to avoid it becoming a loss, he calls a veterinarian to take care of the animal. But if the animals are not for consumption, they kick it out of the house or try to kill it to dispose of it.*

These various values placed on livestock may, in turn, drive or impede prevention behaviors. Participants expressed interest in keeping their animals healthy in order to safeguard their investment. As a woman in rural Kankan explained, *"If you vaccinate them over time, they'll live a long time. They'll reproduce. You'll reap maximum benefit."*

In addition, participants recognized the link between animal and human health, and the benefit of prevention behaviors, such as keeping animals and their enclosures clean. As rural man in Kankan explained, *"If you clean the animals' pen it will reduce your expenses and it allows you to increase animal productivity."* While talking about the behavior of cleaning animal enclosures, a woman in urban Kankan mentioned, *"If you want to profit from your cattle, you have to take care of them. You have to keep the area clean. When you disinfect, you yourself will benefit from good health."* Another participant concurred, remarking, *"First you clean where the animal sleeps, then you clean the animal. So, if these two are clean, and the animal is well cared for, you who eat it will be healthy. But if this place is not clean, you yourself as a human being will not have good health."*

At the same time, the challenges around poverty and demands on providing for one's family may complicate people's ability to keep their animals healthy. For example, because losing animals to illness could mean financial ruin, people may hide sick animals or sell meat that is not safe to eat so as not to incur a loss. An exchange between other urban male participants from Conakry who discussed deciding to kill a sick animal noted:

*Participant 1: When there are no more resources, before the animal becomes so weak that it no longer serves any purpose to people. So people prefer to kill to eat it so that it will at least serve some purpose.*

*Participant 3: Uh. . ., killing the animal when you know it's really sick is hard, especially if it's a cow. If there's no more hope, it's true. But for money, there's more hope, for money. He knows that if the animal dies like that, he's not going to benefit at all, so he anticipates. He kills the animal and then he sells the meat, you know. That's also the impact of poverty.*

In a similar light, a media professional in Conakry summarized:

*. . . Even if you say the cattle there is affected by this, we have to slaughter it so that it doesn't contaminate the others. For them, it's a loss. So we have resistance because the majority of the population is illiterate. They don't directly see the danger of the disease. But, rather their economy, their business, okay! That's what makes people a little reluctant.*

People might also decide to consume meat from a sick animal or from an animal with unknown origin or health history so they can provide nourishment for their family. In a focus group among animal handlers in N'zérékoré, participants discussed the difficulty in convincing people to avoid eating meat from sick animals because people are hungry. They described how people will resort to running away and eating the meat in hiding. One participant claimed, "*It's not easy in the village because there are some families who can go one to two months without eating meat and the children are short of protein and if such an opportunity arises, they will not fail to indulge their hunger.*"

A participant from a focus group of animal handlers in Conakry described the difficulty in deciding whether to consume potentially bad meat,

*It happened to me once during that time I was grilling meat to resell, I was sold a meat that is already bitten by the dog and the veterinarian told me not to resell this meat and I answered why that because I have already invested my money in it and he insisted to bury it I said no I will take it home, he advised me that my family should not eat, I explained that because the dogs capture animals for us to hunt and we eat that and why not a goat?*

*So in the end I thought about it and I said if I take it and the family gets sick I am responsible. I accepted and did what the vet had [said].*

*If not you get rid of him like that it is not easy, because even the people from whom we bought this animal will not refund our money.*

Furthermore, even though people may want to purchase meat that is safe for consumption, it may be out of their financial means to do so, as an animal handler from Kankan explained,

*When you send this sick meat to the market, people will buy it. They won't know what killed the animal. . . since you can't afford to buy the healthy meat, you end up in the situation of buying the unhealthy meat. That's it.*

Because of the monetary value of animals, a related issue that surfaced was fear of theft. When talking about the feasibility of keeping animals separate from human living areas, participants raised how the fear of theft made enacting the behavior more difficult, even though they might be aware that the behavior is beneficial for both human and animal health.

Some individuals talked about the lack of available space for secure penning, although others, especially in Kankan, described their fear of others stealing their prized animals as a reason for believing that this behavior was not feasible to achieve. For example, a man from rural Kankan remarked, "*The fear we have, the risk of being robbed. Otherwise, this practice is inexpensive and is hygienic since it protects us from the smell of animal urine and excrement. Because of the fear of being robbed it is not easy.*"

In direct response, another man in the same focus group agreed, "*What he said is the simple truth. Me, I am obliged to sleep with my calves out of fear of having them stolen. So, for me, this practice is not easy to observe.*" Similarly, an animal handler in Conakry stated, "*If you distance*

*yourself from the sheep or goat pen, they will steal from you at night. As it is raining now, if your goat pen is far away, even if you put sheet metal, they will take it away and you won't know any-thing. That is why it is difficult to keep the goat/sheep at a distance."*

To avoid theft, some people resort to keeping their animals in their compound. A farmer from Kankan stated, *"Nowadays, keeping the animals in the courtyard, better to keep the cattle in your house. They will come and take your cattle and take them away. So keeping the animal away from us right now is hard. It used to be done in the past, but today it's not possible."*

## Discussion

This study provides helpful implications to improve current and inform future programs aimed at the prevention, control, and surveillance of known priority diseases for Guinea and for future outbreaks and emerging zoonoses. In general, the data suggest that although some people may perceive animals as possible sources of disease transmission, they may not take the appropriate preventive precautions. One reason people aware of risks may not engage in prevention behavior was the ongoing lack of certainty of their ability to control their risk. These attitudes stemmed from feelings related to lack of autonomy and ability to control other people's behaviors. For example, people are unable to control whether their neighbors vaccinate their dogs for rabies, and, therefore, even if people adopt preventive behaviors, they cannot fully minimize their risk. In the face of regular uncertainty, participants expressed a desire for consistent, accessible information they could trust. These findings support previous research in Guinea suggesting that during the EVD outbreak people were resistant to adopt new behaviors given the uncertainty around prevailing information sources [16]. Furthermore, research on the COVID-19 pandemic demonstrates how uncertainty acts as a hurdle to successfully implementing preventative behaviors [21]. Similar to some zoonotic outbreaks in Guinea, the COVID-19 pandemic spread quickly and widely making it difficult for individuals to learn, adopt and modify behaviors as new information became available [21, 22]. Increasing the availability of clear and reliable information therefore represents an important step in helping people to overcome perceived inability to control diseases and their associated risks.

Another central reason for not engaging in zoonotic disease prevention behaviors centered around the importance of animals for the livelihood and wellbeing of Guinean families. This finding is in line with previous research on zoonotic disease in other contexts, which identified the high value of animals as a barrier to preventive health behaviors due to the economic impact of those behaviors [23–27]. Consistent with other literature, the current study found that the value individuals ascribe to their animals vary, including protecting the household, providing essential nourishment, and securing financial income [28]. These values may overlap and be somewhat fluid. For example, cattle raised for eventual live sale can rapidly, if sick, become destined for household consumption. While individuals may know that eating a sick animal may present a health risk, prior research suggests that the intention to adopt protective behaviors is motivated by comparative risk perception as much as absolute risk perception. That is, risk is not evaluated by individuals in a cognitive or social silo but involves comparison to the risk of other potential health issues [29]. For certain prevention behaviors explored in this study–keeping animals away from the house, avoiding eating certain types of meat, or burying sick animals–the perceived threat of an uncertain health risk might be insufficient in the face of a more immediate and pressing concern, such as hunger or financial loss. In other words, the health benefit of avoiding possible disease in the future might not outweigh the more urgent need to provide nutrition to their family, resulting in people opting to eat sick animals or meat of uncertain origin. The exception was a situation where a trusted figure, a

veterinarian, prompted deliberation by instructing an individual not to eat meat from an animal that had been bitten by a dog. The participant evaluated the potential economic consequences (hunger and loss of money) against the potential health consequences (family getting sick) and accepted the economic loss. More often, participants discussed different decisions, but likewise emphasized that comparative risk assessment was driving decision-making, not absolute risk perception of the disease itself. Public health professionals, thus, may need to go beyond continually reinforcing the absolute risk of zoonotic diseases but frame zoonotic disease threats in alignment with other threats, as participants in this study noted. Vaccination and keeping pens clean are the examples participants gave, emphasizing that keeping animals clean and vaccinated protects their financial investment. A more convincing way to promote the prevention behaviors of vaccination and keeping pens clean may be to focus more around protecting the health of one's animal to protect their financial investment as opposed to only focusing on reducing their risk for zoonotic disease infection. As some have argued, behavior change interventions in contexts like Guinea cannot ignore structural interventions and economic development, as individual behavior related to animal use is highly constrained by food insecurity and the inability to access capital [30]. Yet the economic argument–that even when people know the right thing to do, they cannot afford to change, or they are not sufficiently motivated in light of more pressing concerns–must also be complemented with an understanding of the sociocultural history of co-habitation between humans and animals in the specific area.

Many of the behaviors discussed in the current study are ones that require an ongoing commitment and would benefit from habit formation. Research on habit formation suggests that sustaining habits calls for repetition of the behavior, relevant cues to action for the particular audience, and tangible rewards [31]. For example, in order to more successfully promote better animal husbandry habits, future programs may want to capitalize on the value that people place on their animals for their own livelihood. Existing habits may also serve as a barrier to introducing a new or modified behavior, as was found in the current study. As a result, future programs may want to proactively seek ways to reward ideal habits and disincentivize negative existing habits.

## Limitations

The current study experienced three limitations. First, the study only included participants from three prefectures. Due to financial and time constraints, the study was unable to extend participation to additional prefectures. At the same time, the study does not claim to be representative of the entire country. Moreover, the selected prefectures represent areas of particular interest for zoonotic disease. Second, the study explored people's perceptions about what their community believes, not whether they themselves actually perform the behaviors. Asking about perceptions of others rather than personal behavior was an intentional design to get people to openly talk about their opinions and beliefs even if not directly disclosing their personal behavior. Third, there were some inconsistencies in the accurate application of the focus group guides, resulting in some participants being asked about behaviors that similar audiences in other prefectures were not. This resulted in extra data on certain behaviors but not for all prefectures, meaning that it was difficult to assess whether trends in the data would have been found across the three prefectures. The insight gained from this additional data was, nevertheless, relevant for those particular audiences and provided useful information. Regardless of these limitations, the current study contributes to the evidence regarding behavior and zoonotic disease prevention in among similar populations in Guinea and the West African region.

### Recommendations and conclusion

Two key recommendations came from the findings of this study. First, policies and programs should work to reduce uncertainty and make it easier for people to safeguard their investment and livelihood found in animals. For example, making meat certification more visible, such as via a seal of approval, could increase people's confidence in the meat they consume. Similarly, the creation of solidarity funds for farmers could help reduce the financial burden of actions such as hiring someone to guard their animals from roaming too far or from theft, as well as help secure funds for vaccination and materials/supplies. Second, as stated above, the study reinforces the importance of communication in the face of uncertainty in the human and animal health environment. Study findings support the continued strengthening of communication channels between the national, regional, and local levels, especially for disease surveillance. Such a programmatic effort could be especially effective given that people place greater trust on messengers from their own communities. For example, it would be important to improve and expand the existing capacity of local stakeholders, such as community health providers, in order to prepare for current and future diseases that might arise in communities. The implementation of these recommendations has the potential to play an important role in improving people's health and well-being in relation to zoonotic disease.

Though the list of priority zoonotic diseases in Guinea continues to evolve, the need to discover ways to effectively promote the related prevention behaviors remains. The current study provides insights into perceptions of zoonotic diseases that are of interest to the Guinean government as well as factors influencing related behaviors. In addition, study findings can inform future policies and programs, whether for current or future priority zoonotic diseases, both in similar populations in Guinea and other countries in West Africa.

## Supporting information

**S1 Appendix. Focus group discussion guides.**
(DOCX)

**S2 Appendix. In-depth interview guides.**
(DOCX)

**S1 Data. GHS PLOS qualitative data.**
(DOCX)

## Acknowledgments

The authors would like to thank Antonia Morzenti, Hannah Mills, and Dr. Mamadou Saliou Barry for their contributions during data collection and analysis. We would also like to thank the team of data collectors and government of Guinea ministry representatives who participated in the data analysis workshop from which the insights in this manuscript came. Finally, we are grateful for the men and women who spoke with us and shared their time and opinions.

## Author Contributions

**Conceptualization:** Tilly A. Gurman, Natalie Tibbels.

**Data curation:** Tilly A. Gurman, Kendela Diallo, Elizabeth Larson, Natalie Tibbels.

**Formal analysis:** Tilly A. Gurman.

**Investigation:** Tilly A. Gurman, Kendela Diallo.

**Methodology:** Tilly A. Gurman, Natalie Tibbels.

**Project administration:** Natalie Tibbels.

**Supervision:** Tilly A. Gurman.

**Validation:** Tilly A. Gurman, Kendela Diallo.

**Visualization:** Kathryn Sugg.

**Writing – original draft:** Tilly A. Gurman, Natalie Tibbels.

**Writing – review & editing:** Kendela Diallo, Elizabeth Larson, Kathryn Sugg, Natalie Tibbels.

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
