## [Decision Letter · Decision Letter 0]

5 Oct 2022

PGPH-D-22-01135

Recurring uncertainty in animal encounters and the importance of animals for one’s livelihood as influences on zoonotic disease prevention behaviors: Findings from a qualitative study in Guinea

Dear Dr. Gurman,

Thank you for submitting your manuscript to PLOS Global Public Health. After careful consideration, we feel that it has merit but does not fully meet PLOS Global Public Health’s publication criteria as it currently stands. Therefore, we invite you to submit a revised version of the manuscript that addresses the points raised during the review process.

We look forward to receiving your revised manuscript.

Kind regards,

Ismail Ayoade Odetokun, DVM, Ph.D.

Academic Editor

Journal Requirements:

a. Please clarify all sources of funding (financial or material support) for your study. List the grants (with grant number) or organizations (with url) that supported your study, including funding received from your institution. 

b. State the initials, alongside each funding source, of each author to receive each grant.

c. State what role the funders took in the study. If the funders had no role in your study, please state: “The funders had no role in study design, data collection and analysis, decision to publish, or preparation of the manuscript.”

d. If any authors received a salary from any of your funders, please state which authors and which funders.

2. We have noticed that you have uploaded Supporting Information files, but you have not included a list of legends. Please add a full list of legends for your Supporting Information files after the references list.

Additional Editor Comments (if provided):

Please, kindly use the comments from the reviewers to revise your manuscript while ensuring that all issues raised are attended to adequately.

Reviewers' comments:

Reviewer's Responses to Questions

**Comments to the Author**

1. Does this manuscript meet PLOS Global Public Health’s publication criteria? Is the manuscript technically sound, and do the data support the conclusions? The manuscript must describe methodologically and ethically rigorous research with conclusions that are appropriately drawn based on the data presented.

Reviewer #1: Yes

Reviewer #2: Partly

2. Has the statistical analysis been performed appropriately and rigorously?

Reviewer #1: I don't know

Reviewer #2: N/A

3. Have the authors made all data underlying the findings in their manuscript fully available (please refer to the Data Availability Statement at the start of the manuscript PDF file)?

Reviewer #1: No

Reviewer #2: Yes

4. Is the manuscript presented in an intelligible fashion and written in standard English?

Reviewer #1: Yes

Reviewer #2: Yes

5. Review Comments to the Author

Reviewer #1: The article identifies two themes related to zoonosis prevention behaviors in Guinea. The study is relevant and warrants further investigation into mitigating factors as identified by this study.

The first theme identifies uncertainty in animal encounters, referring to potential contact with dogs, and also contact with meat. Perhaps "encounters" referring to contact with meat products is not the correct term to use?

In the results section, a diagram depicting how the team arrived at the two main themes showing other subthemes identified in the study would add value to understanding the background of the study.

It is mentioned that observational studies were also done on 15 participants, the results of this part of the study is not discussed or mentioned in the results section.

A few typos:

116 Typo - executives

418 TYpo - improving

Reviewer #2: GENERAL/MAJOR COMMENTS

This is a very well written paper, concerning a subject which is receiving more and more research attention and that is important for improving public and animal health and reducing antimicrobial resistance, and for achieving the related sustainable development goals. In order to warrant publication, however, some major adjustments are needed: A section describing the existing knowledge concerning what factors shape preventive behaviour for infectious diseases, especially in resource strained communities is missing from the introduction. Further needed is a description of the three study areas and the people living there including the local context of public and animal health status and prevention, animal husbandry, and not the least, the poverty situation! It is actually never mentioned in the introduction that Guinee is a low-income country, and how this affects both the resources allocated to public and animal health authorities for preventive work, peoples’ dependance of livestock for their livelihoods, or the actions people are able to take in connection to infectious diseases. This needs to be emphasized. Important information is further missing from the materials and methods-section, making it impossible to verify if the conclusions drawn from the study are valid: the sample selection process need to be described in much more detail as need the interview situation and methodology used during the interviews and focus group discussions, including if any interview protocols were used and how group dynamics and power balance were handed in the focus groups, and how the focus groups were composed (homogenous/heterogenous and considering which factors). If interview guide were used these needs to be annexed. As the article is written in English and quotes are in English, and interviews were translated and transcribed in French, it needs to be stated at what point in the analysis the quotes were translates, and if the translation is verbatim from the transcripts or not. I would further recommend to reduce the number of quotes in the results section, to chose a couple for each novel theme. In the first part of material and methods 3 activities are mentioned; individual interviews, focus group discussions and participant observations. A more detailed description of the methodology (including inclusion/exclusion criteria for sample selection and if any observation protocol was used etc) as well as the results from the participant observations are however missing from the manuscript. Another shortcoming is that the results from some of the analytical steps described in the material and methods are not present in the results section making it difficult to understand how the “novel themes” emerged.

Both the results section (if the quotes are not included) and especially the discussion are short and need to be developed. I think the policy recommendation could be better expressed in for example a policy brief or similar than in a scientific paper, and suggest removing at least part of that section, leaving the conclusions of the study.

DETAILED COMMENTS

Title

Please consider rephrasing the title, I have re-read it many times but still can not quite understand it. Maybe just remove “as” and “on”? And why “recurring”?

Abstract

Line 18-19: Please clarify how/why zoonotic diseases have been at the forefront in Guinea (due to recent outbreaks, or public health priorities or other?)

Line 19-20: This is valid for non-zoonotic diseases as well. Please adjust the sentence accordingly.

Line 22: There seem to be “a bridge” missing between the previous sentence mentioning preventive behaviour and this mentioning risk communication, please clarify the link between these two items.

Line 24: Is the part of sentence after the “-“ meant to summarise all the listed categories of people? In that case maybe better replace the “-“ with “i.e”?

Line 25: According to the material and methods it seemed you first did deductive coding and after let initial themes emerge. Is the latter what you here refer to as inductive coding? Please clarify (put the different coding steps in chronological order) and use the same terms in the different sections.

Line 29: What do you mean by “novel” in this regard? Please clarify.

Line 35: Do you by “infected meat” mean “meat from animals know to be suffering from an infectious disease at the time of slaughter”? “To me “infected meat” sounds like meat that has been purposively infected (spiked) with something. Please rephrase accordingly.

Line 35: Priority for whom? Please clarify.

Line 37: I think “insights (or results) from the study” reads better. Please consider rephrasing.

Introduction

Line 54-57 and all through: Do you refer to the disease in humans or animals here? I think the correct term for the disease and the pandemic in humans is Covid-19, whereas the disease-causing agent is of course SARS CoV-2, which is also the name of the disease in animals (infection with SARS CoV-2). Please clarify.

Line 57: I don’t see the need for the need for the sub-sentence “in consequential”. Please consider rephrasing.

Line 62-63: Do you refer to both high and low pathogenic avian influensa here? Otherwise you can use the abbreviation HPAI that you previously introduced.

Line 65-71: I think the jump in discourse from the role and responsibility of organization (international, national, regional, community based) in disease preventions to that of individuals is very important for the objectives of the study and deserves to be better described. Please develop this section a bit more.

Line 70-73: Please incorporate these lines together with the mentioning of the EVD outbreak in Guinea earlier in the introduction in line 60-63.

Line 79-81: Yes, but a lot has been studied on these exact subjects regarding the EVD outbreak in Sierra Leone (see for example Hewlett and Hewlett 2007, Abramowitz et al 2015, Roca et al 2015). Even if the local situation in Guinea is different enough to justify specific studies on this context (and the studies on EVD don’t include the human-animal interactions), I would recommend including a short section about the insights from the numerous studies on the EVD-outbreak in Wes Africa.

Line 82: What kind of “dearth” do you refer to here? The same kind as in the previous sentence (human-animal inter actions, societal and economic drivers of beahaviour etc) or do you refer to lack of epidemiological data (prevalence, incidence etc)? Please clarify.

Line 84-85: Please clarify what you mean by “priority” (for whom and why and with what purpose) and describe how the list of priory diseases was arrived at (in more details than “in collaboration” and not just by giving the reference to the report).

Line 85: Please write out abbreviations first time used.

Line 86: I am confused what is meant by human anthrax in this regard? Please clarify, including if all prioritized disease were only prioritised by the public health authorities for disease manifestations in humans?

Line 86: See previous comment about LP/HPAI.

Line 90-91: I presume this is a second aim/objective of the study, can you please reformulate accordingly.

Line 91: I am confused to the last part of this sentence (“beyond EVD”), from the introduction there is nothing indicating that study only focuses on EVD? Please clarify.

Materials and methods

Lime 92: I didnt look into the author instructions regarding subsections, but most commonly this section is referred to as “Materials and methods”. Please check this up and change if called for.

Line 95: Please describe how and why these three regions were selected.

Line 98: Should this be in “the three respective regions” or do you presume that the results can be generalised to country level?

Line 103: Please describe the selection process in much more detail, did you for example select study sites within the three selected regions? How? How was the stratification urban-rural mentioned below done?

Line 106: Such interviews with individuals purposefully selected are commonly called key-informant interviews. Were these individuals key informants?

Line 107: As “veterinarians” are mentioned on the next line, I presume these are public health providers?

Line 108: What categories of veterinarians were these? If they were animal health providers, were they all graduated veterinarians (farmers sometimes refer to all animal health providers including community-based animal health workers as “veterinarians”). Please clarify.

Line 108: Please clarify the inclusion/exclusion criteria for “the general public”. Did they need to own/take care of animals/have animal contact for example?

Line 109: I have never seen the expression “animal handlers” in these contexts before (as far as I know the expression is most commonly used for example for someone handling animals at a zoo or a circus). Maybe better use “animal owners or care-takers”? And for the butchers, was that the only category of stakeholders in the livestock value chain apart from farmers that were included? Why? Please clarify the selection criteria and process.

Line 113: As it is mentioned that the research group consisted of both males and females, were gender aspect considered in any specific way during the data collection? Please clarify.

Line 118: Please clarify how individual participants were invited and by whom.

Line 122-123: Did interviews and focus groups follow a topic guide or interview protocol? What was the topic areas covered? What was observed during the observations? Please clarify.

Line 135: Please include some references to the methodology used for the analysis.

Line 138: Please provide a more detailed (manufacturer, country) reference for the software used.

Line 138: Does this mean that all transcripts (of individual interviews and focus groups) were imported to the software? Please clarify.

Line 138-140: If the members of the research team mentioned are among the authors of the paper, please include the initials or author-list order instead.

Line 140: Is the nationality of the senior research relevant for this section?

Line 140-143: Please rephrase these sentences, this step seems to be an important part of the data analysis, but now it seems to be described partly as a training?

Line 144: Was this all transcripts coded according to the initial coding framework mentioned on line 136, or parts of the transcript that fitted the initial coding? Please clarify.

Line 146: Are the “key themes” the same as the “novel themes” discussed in the results section?

Results

Line 149: Please mention how the participants/activities were divided across the three regions as well as min-max participants/focus group.

Line 154: Please verify the sentence for grammar and synthax. I presume these results are from the participant observations, but that is not clear, please clarify this.

Line 155: Are the “novel themes” the same as “key themes” mentioned in the previous section? Information/result about the initial coding framework/codes used, initial themes (are these the same as what is commonly called emerging themes?), and the key themes emerging after the discussions in the plenary discussion of the participatory data analysis workshop is missing. Please add.

Line 155: Please remove “interesting” from the results section, if the results are interesting or not can be mentioned in the discussion.

Line 155-156: This seems to belong to materials and methods/data analysis? Although I don’t understand where in the described data analysis process it was performed. Please clarify.

Table 1: Please clarify (see previous comment) for both the human health and animal providers if they were professionals or community-based health workers. Please specify what is a vendor (of meat or live animals?). Why are hunters and transporters grouped together? See previous comment about “the general population”. What is a “Media professional”? Please provide the gender distribution also for the other respondents apart from the “general population”.

Line 159: I would suggest using the same wording for the novel themes in the result section and in the title. Please consider rephrasing in one of the places.

Line 160-164: Is this a summary of this theme? If not, some information seems to be repeated on line 165 and onwards, please check this.

Line 163: An “and” seems to be missing. Should it be “sufficient for”?

Line 165 and 269; Were the data analysed according to the different regions? If so please describe this in the M&M-section.

Line 177: The first part of this quote seems not to be related to uncertainty but to stakeholder cooperation? Please clarify.

Line 201: Did the participants really advocate “slaughtering”, or rather euthanizing or killing/putting down? Please clarify.

Lnie 201: Please remove “unsurprisingly” from the results and return to such judgments in the discussion, if necessary.

Line 202: Half of the focus groups or half of the participants in a specific focus group? If the fist, please describe in material and methods how consensus was reached.

Line 203: By “health” of the meat, do you mean “safety” (as in food safety), or the health of the animal from which the meat derives? Please clarify.

Line 217-220: Please put the quotation in italics.

Line 254: “Health agent” has not been mentioned as a category of participants before. Please clarify.

Line 268: In the title this theme is called “Importance of animals for”, which I believe is correct. Please align throughout the manuscript.

Line 286: Some parts of the text which seems not to be a quote are in italics, please check this.

Line 295: Please see previous comment regarding using the term “infected meat”.

Line 312: This sentence reads like any kind of bushmeat poses the same threat concerning zoonotic diseases as meat from sick livestock, which I find controversial. Please clarify.

Line 313: Please change to “a focus group”.

Line 331 (and other places): Along the same lines as for “infected meat”, do you mean “meat from healthy animals” or meat that is specifically healthy/beneficial for the health? Please clarify.

Line 332: For some quotes you indicate the gender of the respondent, for some not, please be consistent in this matter.

Discussion

Line 357: What do you mean by “shed light on possible implications”? Please clarify this sentence.

Line 358: Are “the cross-cutting themes” the same as “the novel themes” mentioned in the results, or as the “key themes” mentioned in materials and methods, or something else (what in that case?)? Please clarify.

Line 358-360. What do you mean by “helpful for the priority diseases”? Helpful in what way? For improving control? Please clarify.

Line 360: What do you mean by “future zoonotic diseases”? Emerging zoonoses? Or future outbreak of zoonotic diseases? Please clarify.

Line 361-363: What about the data from the participant observations? Please clarify.

Line 365-367: Please don’t repeat results in the discussion.

Line 376: Please remove an abundant “information of”.

Line 379 and 422: Given the restricted geographical coverage and skewed gender balance in the sampled population, please consider if your results are representative for all Guinean families/in Guinea.

Line 379-383 and in other places: Please consider the use of the word “behaviour”. In some instances I think you rather refer to for example “preventive actions”. Especially constructions such as “to behave” can easily read as the researcher taking a top-down position.

Line 379: I think the sentence would read nice if you replaced “outside Guinea” with for example “in other contexts (or settings or countries).

Line 387: “Own” is not needed in this construction.

Line 388: Please remove “likewise” and reformulate the sentence.

Line 395: I agree, please see my comment regarding the claims made on line 379 and 422 in this regard.

Line 397-399: This are very interesting methodological aspects, please include them in the material and methods. I also get a bit confused as you in the result section several times refer to differences in expressed knowledge of preventive actions and behaviour. Please clarify in both sections.

Line 398: Which study component? Please clarify.

Line 400: Please describe the intended use of the focus group guides (and that you had any! And annex them) in the materials and methods section, and describe how you dealt with the problem you describe here. Otherwise the validity of the data can not be assessed.

Line 401-404: It seems focus groups included participants being asked about certain behaviours. This is also new, and very interesting information that needs to be given in the materials and methods section.

Line 401: What do you mean by “intended audience”? Focus group participants? Please clarify.

Line 408: I am a bit hesitant to include (policy) recommendation in a research paper. I would find that more fitting in a separate policy brief (based on the research at hand). Please consider changing this section.

Line 410: What do you mean by “ecosystem” here? I can not see that the paper is about ecosystems?

Line 410: Is it really possible to remove uncertainties in the livelihoods for most (poor) Guineans? I think “reduce” would be more fitting in this regard.

Line 414; I believe that communication is always multi-(or at least bi-) directional, otherwise it is simply information. Please consider rephrasing.

Line 415. Two “support” on the same line, please rephrase the sentence.

Line 416: “Of the effectiveness” seems redundant in this sentence. I am also not entirely convinced that the data supports this statement. Please clarify.

Line 420: See previous comment about “priority zoonosises”.

Line 421: What do you mean by “multiple related”? Do you refer to that hygiene or other preventive measures usually protect against several infectious diseases? Please clarify.

Line 423: Of interest for whom? Please clarify.

Line 423-427: This is already said on line 357-358. Again also consider how the results are representative.

Line 425: What do you mean by “helpful”? Not the same as in the sentence on line 423-425? Please clarify.

6. PLOS authors have the option to publish the peer review history of their article (what does this mean?). If published, this will include your full peer review and any attached files.

**Do you want your identity to be public for this peer review?** For information about this choice, including consent withdrawal, please see our Privacy Policy.

Reviewer #1: **Yes: **Ilana van Wyk, University of Pretoria

Reviewer #2: **Yes: **Erika Chenais

---

## [Decision Letter · Decision Letter 1]

13 Feb 2023

PGPH-D-22-01135R1

How do the recurring uncertainty and importance of animals for one’s livelihood influence zoonotic disease prevention behaviors? Findings from a qualitative study in Guinea

Dear Dr. Gurman,

Thank you for submitting your manuscript to PLOS Global Public Health. After careful consideration, we feel that it has merit but does not fully meet PLOS Global Public Health’s publication criteria as it currently stands. Therefore, we invite you to submit a revised version of the manuscript that addresses the points raised during the review process.

We look forward to receiving your revised manuscript.

Kind regards,

Ismail Ayoade Odetokun, DVM, Ph.D.

Academic Editor

Journal Requirements:

1. We have noticed that you have uploaded Supporting Information files, but you have not included a list of legends. Please add a full list of legends for your Supporting Information files after the references list.

Additional Editor Comments (if provided):

Reviewers' comments:

Reviewer's Responses to Questions

**Comments to the Author**

1. If the authors have adequately addressed your comments raised in a previous round of review and you feel that this manuscript is now acceptable for publication, you may indicate that here to bypass the “Comments to the Author” section, enter your conflict of interest statement in the “Confidential to Editor” section, and submit your "Accept" recommendation.

Reviewer #1: All comments have been addressed

Reviewer #2: (No Response)

2. Does this manuscript meet PLOS Global Public Health’s publication criteria? Is the manuscript technically sound, and do the data support the conclusions? The manuscript must describe methodologically and ethically rigorous research with conclusions that are appropriately drawn based on the data presented.

Reviewer #1: Yes

Reviewer #2: Yes

3. Has the statistical analysis been performed appropriately and rigorously?

Reviewer #1: I don't know

Reviewer #2: Yes

4. Have the authors made all data underlying the findings in their manuscript fully available (please refer to the Data Availability Statement at the start of the manuscript PDF file)?

Reviewer #1: No

Reviewer #2: Yes

5. Is the manuscript presented in an intelligible fashion and written in standard English?

Reviewer #1: Yes

Reviewer #2: Yes

6. Review Comments to the Author

Reviewer #1: Availability of data should be addressed.

Reviewer #2: MAJOR COMMENTS

As I pointed out in the first review, the result section is over-crowded with quotes. You need to remove at least half of these (I have pointed out some examples of quotes illustrating similar themes, but not for all instances). Instead of (almost) only displaying the results as quotes you need to develop your themes (main themes and cross-cutting themes) and the analysis according to the framework you describe.

A bit in a similar way (and as I also pointed out in the first review) the discussion sections is to short and to some extent shallow, and needs to be developed, in particular referring to how you discuss your results against the existing literature (instead of repeating results). A big part of the already short discussion is further policy recommendation (starting already at the end of section just before “limitations”). You further need to discuss also the limitations of the study against the existing literature. Regarding the issue of including policy recommendation in a scientific article, I support the argument that one of the most important tasks of research is to contribute to policy change, I however insist that this is better done is separate fora, each adopted to the intended purpose. You can rewrite the recommendation section and include much of the same information, but in the form of a discussion, appropriate for a scientific journal.

Detailed comments:

Title: The revised title reeds better but is still quite a mouth-full, especially having part about “uncertainties” and “importance” in the same sentence. I would recommend another try at finding a more fitting title.

Abstract

Line 18: you mentioned in a response to one of my queries on the first version of this manuscript that you were referring to highly pathogenic avian influenza. Please change all through the manuscript.

Line 24: (and throughout the manuscript): I read your explanation about using the term “animal handlers” including the translation from the French word used. If you think the French term is important, I suggest you include it in brackets or as a foot note. For English, I think for example “people taking care of/looking after animal/interacting with animals” or herders (if that is what they were) are much more common, and better. If referring to several different categories (herders, hunters, middle men, trader, butchers) I recommend using several different terms.

Line 29: From your additional description of the methodology, this doesn’t seem to be a correct description? Didn’t 7 themes arose, butarise, and you choose to explore 2 (the most novel)?

Line 33-35: Here you mention “vaccinating animals” and “selling unsafe meat” as example of preventive behaviors. In order for that to make sense you need to explain your definition of “prevention”, especially in connection to infectious diseases, or choose another wording (referreferring to the term and definition of “preventive medicine”). The first is of course a classic preventive measure, protecting herds against infectious diseases. Do you mean here that “selling unsafe meat” is a preventing behavior as it protects the farmer from a possible financial loss if not selling meat from infected animals? That is of course true, bit grouped together with a traditional infectious disease preventive measure like that it needs explanation/rephrasing.

Introduction

Line 47: “2,5 billion illnesses” is difficult to grasp, it is more useful to measure the impact in DALYS. I believe some more recent references with impact might be available.

Line 48-59: Please check how you capitalize disease names, and be consistent through the manuscript. Given as this is not a popular scientific paper I would recommend removing “bird flu” and “mad cow disease”.

Line 51: There seems to be either a “(“ missing or a ”)” to many.

Line 77: Do you mean “at” multiple levels?

Line 84: I think a more frequently used term is “hens, chickensDo you mean “poultry” or maybe “poultry and other captive birds”?

Line 89: Do you mean “modern slaughter houses” or maybe “butcheries”?

Line 94-95: Please rephrase this sentence.

Line 96: Do you mean “are” prevalent?

Line 97-99: Please check this sentence, seems to be both surplus words and words missing.

Line 111-115: The sentence seems to be missing an item – “points to xx and yy” for what or as examples of?

Line 125: I would recommend to remove “priority” from this sentence, it doesn’t add any extra information (as the diseases are already listed), but on the contrary adds confusion as to for whom they are prioritized.

Line 126-128: Please reformulate this sentence, for example starting with “and additional aims was to…”

Materials

Line 134-136: Please provide a reference for the assessment of elevated risks.

Line 143: What do mean by “pile-sorting”? please explain and/or give a reference.

Line 144: I think “feasibility” is enough for defining the first dimension (self-efficacy doesn’t explain anything further for me). Pease consider removing.

Line 159: Please include information on selection of study sites (prefectures?) in the three study regions. Unless all prefectures were included. Maybe add addition in the section about the study areas about how many prefectures each region have.

Line 168: Do you mean in “each prefecture of the selected regions” or in “each selected prefecture”?

Line 169, 171 and 172: The recruitment process is not clear, “the general population” is mentioned twice, please clarify.

Line 172: The age-inclusion criteria is already mentioned on line 159. Please remove in one of the places.

Line 185-187: Please cross-check this with how the recruitment is described on line 169 and keep the information to one place.

Line 189: Do you mean “convenient for the participants”?

Line 194: Please annex the interview and topic guides.

Line 221-222: I don’t think the passage about the researchers experience is necessary here. Please remove.

Line 224: Should one more author be indicated here as “study managers” are in plural? Better to indicate the initial first time mentioned a couple of lines earlier.

Line 228: What do you mean with “by region”? That the participant representing one region formed a group, or that each groups worked with data from one of the three regions, or both of these?

Line 232: Do you mean “saturation was reached”?

Results

Line 235: See previous comment about selection of study sites within the regions. Or did you choose participants from all over the regions?

Line 236 -238: Please write numbers smaller and equal to eleven in letters.

Line 240: This seems to be the end of a first section about number of participants. Please indicate the start of a new section. Somewhere around here it would be nice to list, maybe in a table, all themes (all 7), the 2 you choose to include in the article and the cross-cutting themes.

Line 249: A blank line seems to be missing under the table.

Line 267-269: I think this quote is representative of the four quotes on line 260-269 and 281-283, please keep only one.

Line 270: A “that” seems to be missing in this sentence.

Line 293-308: These 3 quotes seem to allure to the same theme, please keep only one.

Line 320-325: These 3 quotes seem to allure to the same theme, please keep only one.

Line 332-339: These 2 quotes seem to allure to the same theme, please keep only one.

Line 369-380: These quotes seem to allure to the same theme, please keep only one.

Line 360-366, 386-399, 422-424: These quotes seem to allure to the same theme, please keep only one.

Line 404-419: These quotes seem to allure to the same theme, please keep only one.

Line 432-444: These quotes seem to allure to the same theme, please keep only one.

Discussion

447-448: Please either remove “presented above” or replace with naming the two themes.

Line 446-450: These two sentences seems repeatative. Please rephrase, or maybe remove the first sentence.

Line 451: Sufficient to say “the data” without repeating the methodology.

Line 453: Was “unpredictability of disease control” a theme/a result? Please clarify this.

7. PLOS authors have the option to publish the peer review history of their article (what does this mean?). If published, this will include your full peer review and any attached files.

**Do you want your identity to be public for this peer review?** For information about this choice, including consent withdrawal, please see our Privacy Policy.

Reviewer #1: No

Reviewer #2: **Yes: **Erika Chenais

---

## [Decision Letter · Decision Letter 2]

5 Jun 2023

PGPH-D-22-01135R2

Balancing the uncertain and unpredictable nature of possible zoonotic disease transmission with the value placed on animals: Findings from a qualitative study in Guinea

Dear Dr. Gurman,

Thank you for submitting your manuscript to PLOS Global Public Health. After careful consideration, we feel that it has merit but does not fully meet PLOS Global Public Health’s publication criteria as it currently stands. Therefore, we invite you to submit a revised version of the manuscript that addresses the points raised during the review process.

We look forward to receiving your revised manuscript.

Kind regards,

Ismail Ayoade Odetokun, DVM, Ph.D.

Academic Editor

Journal Requirements:

1. Please ensure that the Title in your manuscript file and the Title provided in your online submission form are the same.

Additional Editor Comments (if provided):

Reviewers' comments:

Reviewer's Responses to Questions

**Comments to the Author**

1. If the authors have adequately addressed your comments raised in a previous round of review and you feel that this manuscript is now acceptable for publication, you may indicate that here to bypass the “Comments to the Author” section, enter your conflict of interest statement in the “Confidential to Editor” section, and submit your "Accept" recommendation.

Reviewer #1: All comments have been addressed

Reviewer #2: (No Response)

2. Does this manuscript meet PLOS Global Public Health’s publication criteria? Is the manuscript technically sound, and do the data support the conclusions? The manuscript must describe methodologically and ethically rigorous research with conclusions that are appropriately drawn based on the data presented.

Reviewer #1: Yes

Reviewer #2: Yes

3. Has the statistical analysis been performed appropriately and rigorously?

Reviewer #1: Yes

Reviewer #2: Yes

4. Have the authors made all data underlying the findings in their manuscript fully available (please refer to the Data Availability Statement at the start of the manuscript PDF file)?

Reviewer #1: Yes

Reviewer #2: No

5. Is the manuscript presented in an intelligible fashion and written in standard English?

Reviewer #1: Yes

Reviewer #2: No

6. Review Comments to the Author

Reviewer #1: comments were attached

Reviewer #2: General comments:

The manuscript has unfortunately still not reached the quality expected for publication in PGPH, and require another major/moderate revision.

Many of my previous comments have been addressed in a satisfactory way, and the manuscript has improved in clarity. Some previous comments are mentioned in the rebuttal letters as taken care of while this seems not to be reflected in the text. Some changes are reflected in the tracked version but not in the clear version. Other minor issues have arisen as the manuscript has been redrafted. Quite a few of those are of language editorial or just editorial matter and could easily have been avoided by a thorough language/editing check.

Major comments:

Regarding my previous comment 1, 36-43 and 46 that the authours have chosen to ignore. I am obviously aware of the basics of qualitative research methods (otherwise I would (hopefully) not be reviewing this manuscript). I am also aware about different “schools” for how to use quotes. I am not asking for the quotes to be removed, but in the (many) instances there you have several (sometimes up to four) quotes supporting the same theme or the same statement, reduce the numbers.

Regarding my previous comment 2: The authours has done a slight rework of the discussion, but it is still very short and as I mentioned in my previous comment, neither the limitations nor the conclusion is discussed against the current literature. As I recommended, most of the conclusions could be worked into the discussion. There are a lot of literature discussing the matters raised in first part of the conclusion (lines 499-514, barriers to implementation of prevention beahaviour, the relations between livelihood and disease prevention, communication) that could be used.

Minor comments:

Please make sure that figure captions are “stand alone”.

The new suggested title (“Balancing the uncertain and unpredictable nature of possible zoonotic disease

transmission with the value placed on animals: Findings from a qualitative study in Guinea” as per the title page) reads much better. The change does however not seem to be reflected in the manuscript.

Abstract:

Line 25: See my previous comment on “animal handlers” (comment 5 in your numbering of my comments on the previous version), it does not seem that the clarification you mention in your answer is reflected in the text?

Line 33-35: “The value placed on livestock may, in turn, drive and impede prevention behaviors such as vaccinating animals or selling unsafe meat”.

See my previous comment regarding “selling unsafe meat” as a prevention behaviour (comment 7 in your numbering of my comments on the previous version).

Introduction:

Line 59: See comment 9 in your numbering of my comments on the previous version regarding capitalization of disease names.

Line 66-67: The sentences seem to more factors that could prevent disease transmission (vaccination, hygiene practices, care-seeking for zoonotic diseases, or how people protect themselves) than “individual behaviors that can lead to zoonotic events or zoonotic disease transmission” Please clarify this.

Line 78: The sentence would read better with “in recent years” placed at the end of the sentence.

Line 84: See my previous comment (comment 12 in your numbering of my comments on the previous version) on the use of “fowl”.

Line 86: “for transmission of zoonotic diseases” seems to be missing after “risk factor”.

Line 92 and 96: I presume you mean that the mentioned diseases are prevalent in animals? Please specify. It would be good to include a reference for the claims of high prevalence of the zoonotic diseases. I am very aware of the difficulties with official statistic form these kinds of contexts, but I’m sure that your claim is based on something (unpublished data, personal communication, unofficial disease reports)! When the prevalence is unknown it is also sometimes better to use “frequently occurring” or some other wording that is less epidemiologically loaded.

Line 107: The construction “non-bush meat” is innovative, but I still suggest for example “other meat than bushmeat”. Please consider this.

Line 112: Please consider exchanging “cross-cutting” with “general” or “basic”.

Methods:

Line 134-137: Please move this info to the section in the introduction which describes the respective areas.

Line 143: I’m still confused about the pile-sorting. Is it different from “proportional piling” (which is a common tool in participatory epidemiology)?

Line 155: Please rephrase particularly the second part of the sentence to improve grammar and synthax. Please change to “working day”.

Line 170-172: Please rephrase the sentence to improve grammar and synthax.

Line 158 (study sample): Please see comment 21 and 22 in your numbering of my comments on the previous version. Within the 3 selected prefectures, did you not do any further section of study sites before making the purposive selection of participants (you mention local community/local neighborhoods)? I.e did your 229 participants include people from the entire prefectures or only specific sites (I don’t know what is the next lower levels of administrative units, maybe sub-prefecture, village, ward or something similar)?

Line 168: The previous part of this section seems to concern the different methods, and here the subject of selction/mobilizing start. It might help the reader to include a blank line here to mark that this isa new topic.

Line 179: “multi-day” sounds peculiar, please consider rephrasing.

Line 180: There seems to e an abundant “the” in the phrase.

Line 181: Please include a reference to the data collection tools that you have annexed in supporting information (interview, FGD and observation guides) at for example this point.

Line 183-84: Please rephrase the sentence to improve grammar and synthax.

Line 185-188: This sentence seems to refer to selection of the study sample? Please move to the relevant section (approx. line 168-172).

Line 189: See my previous comment (comment 26 in your numbering of my comments on the previous version), it does not seem that the clarification you mention in your answer is reflected in the text?

Line 216: I presume the “research team” are all authors? If the frame work was developed by TG and NT (and not the entire research team) it seems the sentence should start “TG and NT developed….”.

Line 218: Who are these study managers? Co-authours? Please clarify.

Line 221: See my previous comment (comment 28 in your numbering of my comments on the previous version), it does not seem that the clarification you mention in your answer is reflected in the text?

Line 223: Who were the participants in this analysis that were not data-collectors and how were they selected (not mentioned in sample seletion as far as I can see). Please clarify.

Line 231-232: Please rephrase the sentence to improve grammar and synthax.

Results:

Line 234-247: Please format this section correctly and according to author guidelines.

Line 236-38: Please see my previous comment regarding writing out numbers with letters (comment 33 in your numbering of my comments on the previous version), it does not seem that the clarification you mention in your answer is reflected in the text?

Line 243: Do you mean “health prevention behaviour”?

Line 246: I would suggest to start the sentence with "The authours”.

Line 293, 373: There seems to be an abundant [ ]?

Discussion:

Line 446-450: The first two sentences seem to convey very similar messages. Please consider rephrasing.

Line 463: A “the” seems to be missing in front of “Covid-19”.

Line 465: Please rephrase the sentence to avoid the construction with “information of information”.

Line 507: There seems to be an abundant “b”.

Line 518: Please rephrase the sentence to improve grammar and synthax.

7. PLOS authors have the option to publish the peer review history of their article (what does this mean?). If published, this will include your full peer review and any attached files.

**Do you want your identity to be public for this peer review?** For information about this choice, including consent withdrawal, please see our Privacy Policy.

Reviewer #1: No

Reviewer #2: **Yes: **Erika Chenais

---

## [Decision Letter · Decision Letter 3]

20 Sep 2023

PGPH-D-22-01135R3

Balancing the uncertain and unpredictable nature of possible zoonotic disease transmission with the value placed on animals: Findings from a qualitative study in Guinea

Dear Dr. Gurman,

Thank you for submitting your manuscript to PLOS Global Public Health. After careful consideration, we feel that it has merit but does not fully meet PLOS Global Public Health’s publication criteria as it currently stands. Therefore, we invite you to submit a revised version of the manuscript that addresses the points raised during the review process. Please submit your revised manuscript by Oct 20 2023 11:59PM. If you will need more time than this to complete your revisions, please reply to this message or contact the journal office at globalpubhealth@plos.org. Please include the following items when submitting your revised manuscript:

We look forward to receiving your revised manuscript.

Kind regards,

Ismail Ayoade Odetokun, DVM, Ph.D.

Academic Editor

Journal Requirements:

Additional Editor Comments (if provided):

Reviewers' comments:

Reviewer's Responses to Questions

**Comments to the Author**

1. If the authors have adequately addressed your comments raised in a previous round of review and you feel that this manuscript is now acceptable for publication, you may indicate that here to bypass the “Comments to the Author” section, enter your conflict of interest statement in the “Confidential to Editor” section, and submit your "Accept" recommendation.

Reviewer #1: All comments have been addressed

Reviewer #2: All comments have been addressed

2. Does this manuscript meet PLOS Global Public Health’s publication criteria? Is the manuscript technically sound, and do the data support the conclusions? The manuscript must describe methodologically and ethically rigorous research with conclusions that are appropriately drawn based on the data presented.

Reviewer #1: Yes

Reviewer #2: (No Response)

3. Has the statistical analysis been performed appropriately and rigorously?

Reviewer #1: I don't know

Reviewer #2: N/A

4. Have the authors made all data underlying the findings in their manuscript fully available (please refer to the Data Availability Statement at the start of the manuscript PDF file)?

Reviewer #1: Yes

Reviewer #2: No

5. Is the manuscript presented in an intelligible fashion and written in standard English?

Reviewer #1: No

Reviewer #2: Yes

6. Review Comments to the Author

Reviewer #1: line 128 "diseases, the frequency" - add the space after ,

line 180 remove one "."

line 188 "participated in a training [session?]"

line 197 space after "."

line 211 place "." after ")"

line437 "that is safe"

line 552 - font?

line 568 double ..

Reviewer #2: With the last revision the manuscript has been greatly improved, and is now acceptable for publication. I have a few editorial comments and one other comment that I would prefer that the authors see to before proceeding, I however do not need to see the manuscript again.

Editorial comments:

Line 52; There seems to be an abundant comma after “(HPAI)”.

Line 180, line 568. There seems to be an abundant full stop.

Line 197: A space seems to be missing.

Other comment:

Line 86-105: I understand the lack of official data, but please try to find some references (grey literature, oral communication) for at least some of the important background statements in this section.

7. PLOS authors have the option to publish the peer review history of their article (what does this mean?). If published, this will include your full peer review and any attached files.

**Do you want your identity to be public for this peer review?** For information about this choice, including consent withdrawal, please see our Privacy Policy.

Reviewer #1: No

Reviewer #2: **Yes: **Erika Chenais

---

## [Decision Letter · Decision Letter 4]

21 Dec 2023

Balancing the uncertain and unpredictable nature of possible zoonotic disease transmission with the value placed on animals: Findings from a qualitative study in Guinea

PGPH-D-22-01135R4

Dear Research and Evaluation Officer Gurman,

We are pleased to inform you that your manuscript 'Balancing the uncertain and unpredictable nature of possible zoonotic disease transmission with the value placed on animals: Findings from a qualitative study in Guinea' has been provisionally accepted for publication in PLOS Global Public Health.

Best regards,

Ismail Ayoade Odetokun, DVM, Ph.D.

Academic Editor

Reviewer Comments (if any, and for reference):

Reviewer's Responses to Questions

**Comments to the Author**

1. If the authors have adequately addressed your comments raised in a previous round of review and you feel that this manuscript is now acceptable for publication, you may indicate that here to bypass the “Comments to the Author” section, enter your conflict of interest statement in the “Confidential to Editor” section, and submit your "Accept" recommendation.

Reviewer #2: All comments have been addressed

2. Does this manuscript meet PLOS Global Public Health’s publication criteria? Is the manuscript technically sound, and do the data support the conclusions? The manuscript must describe methodologically and ethically rigorous research with conclusions that are appropriately drawn based on the data presented.

Reviewer #2: Yes

3. Has the statistical analysis been performed appropriately and rigorously?

Reviewer #2: Yes

4. Have the authors made all data underlying the findings in their manuscript fully available (please refer to the Data Availability Statement at the start of the manuscript PDF file)?

Reviewer #2: Yes

5. Is the manuscript presented in an intelligible fashion and written in standard English?

Reviewer #2: Yes

6. Review Comments to the Author

Reviewer #2: No further comments, all my commentsor questions have been adressed.

7. PLOS authors have the option to publish the peer review history of their article (what does this mean?). If published, this will include your full peer review and any attached files.

**Do you want your identity to be public for this peer review?** For information about this choice, including consent withdrawal, please see our Privacy Policy.

Reviewer #2: **Yes: **Erika Chenais
